# Characterization of Nuclear and Mitochondrial Genomes of Two Tobacco Endophytic Fungi *Leptosphaerulina chartarum* and *Curvularia trifolii* and Their Contributions to Phylogenetic Implications in the Pleosporales

**DOI:** 10.3390/ijms21072461

**Published:** 2020-04-02

**Authors:** Xiao-Long Yuan, Min Cao, Guo-Ming Shen, Huai-Bao Zhang, Yong-Mei Du, Zhong-Feng Zhang, Qian Li, Jia-Ming Gao, Lin Xue, Zhi-Peng Wang, Peng Zhang

**Affiliations:** 1Tobacco Research Institute of Chinese Academy of Agricultural Sciences, Qingdao 266109, China; yuanxiaolong@caas.cn (X.-L.Y.); shenguoming@caas.cn (G.-M.S.); zhanghuaibao@caas.cn (H.-B.Z.); duyongmei@caas.cn (Y.-M.D.); zhangzhongfeng@caas.cn (Z.-F.Z.); 2Marine Science and Engineering College, Qingdao Agricultural University, Qingdao 266109, China; caominjiyou@163.com; 3Nanyang Tobacco Group Co., Ltd., Nanyang 473000, China; liqian0532@163.com; 4Hubei Provincial Tobacco Company of China National Tobacco Corporation, Wuhan 430000, China; gaojiaming0532@163.com; 5Wannan Tobacco Group Co., Ltd., Xuancheng 242000, China; xuelin0532@163.com

**Keywords:** endophytic fungi of tobacco, nuclear genes, mitogenome, phylogeny

## Abstract

The symbiont endophytic fungi in tobacco are highly diverse and difficult to classify. Here, we sequenced the genomes of *Curvularia trifolii* and *Leptosphaerulina chartarum* isolated from tobacco plants. Finally, 41.68 Mb and 37.95 Mb nuclear genomes were sequenced for *C. trifolii* and *L. chartarum* with the scaffold N50, accounting for 638.94 Kb and 284.12 Kb, respectively. Meanwhile, we obtained 68,926 bp and 59,100 bp for their mitochondrial genomes. To more accurately classify *C. trifolii* and *L. chartarum*, we extracted seven nuclear genes and 12 mitochondrial genes from these two genomes and their closely related species. The genes were then used for calculation of evolutionary rates and for phylogenetic analysis. Results showed that it was difficult to achieve consistent results using a single gene due to their different evolutionary rates, while the phylogenetic trees obtained by combining datasets showed stable topologies. It is, therefore, more accurate to construct phylogenetic relationships for endophytic fungi based on multi-gene datasets. This study provides new insights into the distribution and characteristics of endophytic fungi in tobacco.

## 1. Introduction

Studies examining fungal endophytes in tobacco plants have shown that they are widely distributed in nearly all tissues and that they play vital roles in tobacco biology due to their various biological functions [1,2,3]. It is therefore important to determine the distribution of endophytic fungi species and their characteristics in tobacco. Pleosporales, the largest order in the fungal class Dothideomycetes, includes a large number of species, including saprobes, parasites, epiphytes, and endophytes [4]. Our previous studies have shown that many endophytic fungi belonging to the Pleosporales order can be isolated from the healthy tissues of tobacco (*Nicotiana tabacum* L.) [5]. The family Pleosporaceae includes species with a variety of ecologies, as plant pathogens, animal parasites, and saprotrophs. Two particularly important parasitic species are *Leptosphaerulina chartarum*, the causal agent of leaf spot disease, which has negative effects on crop yield globally, and *Curvularia trifolii*, which causes leaf blight and leaf spot on different types of feed crops [6]. Herein, using healthy tobacco tissues as the source, we isolated and identified these two important parasitic species.

Recent studies on *L. chartarum* and *C. trifolii* have focused primarily on their pathogenicity. For example, researchers found that *L. chartarum* has a broad host range, generally confining its infections to *Triticum aestivum*, *Miscanthus giganteus*, *Withania somnifera*, *Bromus inermis,* and *Panicum virgatum cv. Alamo*, etc. [7,8,9,10]. However, of note, the fruiting bodies of this fungus have also been reported to cause facial eczema in some animals (i.e., sheep, cattle, goats, and deer) as a consequence of the liver damage caused by a mycotoxin (sporidesmin) [11,12,13]. Several studies have also described the presence of *L. chartarum* conidia in indoor air environments where asthma patients reside [14,15]. *C. trifolii*, a widely distributed, high-temperature pathogen, causing leaf spot on *Trifolium alexandrinum* (Berseem clover) and leaf blight on *Agrostis palustris* (creeping bentgrass) [16,17,18]. Moreover, *C. trifolii* can produce various secondary metabolites in culture, among which, some exhibited excellent anti-inflammatory and antitumor activity [19,20]. Previously, the classification of *L. chartarum* and *C. trifolii,* as well as other species in the order Pleosporales, was determined based on their morphologies [19,21,22]. However, the morphological features of species in this order, including the formation of conidia on conidiophores, as well as the size, shape, color, etc., are easily influenced by environmental factors. Thus, it is difficult to perform classification based on their morphologies alone. Recently, single genes, including 28S and 18S were selected for inferring the phylogenetic relationships between species in the order Pleosporales [23]. Currently, the phylogenetics inferred from combined nuclear genes or organellar genomes, including plastid and mitochondrial genomes, are regarded as effective candidates for phylogenetic relationship analysis due to their moderate variation rate [24,25]. In fact, combining multiple nuclear genes can effectively reduce, or even eliminate, the difference in the rate of mutation of different genes, thereby increasing the accuracy associated with inferring phylogenetic relationships. In addition, the high copy numbers and conserved characteristics in organellar genomes make them readily available and suitable for construction of evolutionary relationships. Meanwhile, employing the whole plastid or mitochondrial genomes can also serve to eliminate the bias in evolution rates of an individual gene [26]. Until now, no studies have reported on the nuclear and mitochondrial genome of *L. chartarum* and *C. trifolii.* Therefore, it is urgent to confirm their genome characteristics and their evolutionary positions before we proceed with future research.

Previous studies have demonstrated that the interaction of plant endophytic fungi may be beneficial for plant growth and disease control [27]. In the process of agelong coevolution with host plants, endophytic fungi could produce various secondary metabolites with kinds of biological activities to assist tobacco in resisting biotic or abiotic stress, promote tobacco growth, and improve their bacteriostatic activities [28]. With the rapid development of modern analytical chemistry and bioinformatics, increasing secondary metabolites from *C. trifolii* and *L. chartarum* with excellent activities were exploited [29,30]. Currently, two endophityc fungi *C. trifolii* and *L. chartarum* were isolated from tobacco. Therefore, these two endophityc fungi in tobacco are good candidates for digging their secondary metabolites, which can be used to assist tobacco growth, also to help tobacco tissue resist biotic or abiotic stress. Global understanding of these two endophytic fungi genomes and their evolutionary histories can help us understand these characteristics, and aid to explore the secondary metabolites exploration in tobacco.

With the rapid development of new sequencing technologies, the nuclear and mitogenomes of fungi have improved tremendously in recent years. Hundreds of available genomes in the public databases have provided an efficient resource for their classification, and analysis of their phylogenetic relationships. However, only a small number of sequences have been published for Pleosporales spp including those belonging to the class Dothideomycetes [31,32,33]. Herein, we sequenced the complete nuclear and mitochondrial genomes of *L. chartarum* and *C. trifolii*. Furthermore, we analyzed and characterized the evolution and structural organization of the *L. chartarum* and *C. trifolii* genomes and inferred their phylogenetic relationships using the mitogenomes and combined nuclear genes. The current results may serve to further the current understanding regarding the genomic characteristics and evolutionary histories of Pleosporales species.

## 2. Results

### 2.1. Fungal Isolation and Identification

The strains studied were identified as *C. trifolii* and *L. chartarum* on the basis of their morphologies and ITS region sequences. Blast results of the ITS1 and ITS4 of *C. trifolii* showed 99.64% and 99.82% homologies with sequences from *C. trifolii* isolate FUNBIO-2 (GenBank Accession No. KC415610.1) and the *C. trifolii* strain AL9m5_2 (GenBank Accession No. KJ188716.1), respectively. Similarly, the *L. chartarum* was identified based on both ITS1 and ITS4 sequences. The ITS1 showed 99.73% homology with that of the *L. chartarum* strain with the GeneBank Accession No.KM877491.1, and the ITS4 sequence was found to possess 100% homology with that of the *L. chartarum* strain KNU14-16 (GenBank Accession No. KP055597.1) (Figure 1) [23]. Our results from the phylogenetic tree analysis supported these alignment results. For example, the phylogenomic investigation of the ITS1 sequence demonstrated that the strain was *C. trifolii*. In our analysis, the phylogenetic relationships based on ITS1 also showed that the *L. chartarum* strain had a high bootstrap value of 99% (Figure 1) [23].

### 2.2. Nuclear Genome Features of L. chartarum and C. trifolii

Next, to understand the features of *L. chartarum* and *C. trifolii*, we sequenced and assembled their genomes. In total, 2.52 Gb and 2.47 Gb of clean reads were sequenced by Illumina HiSeq platform for *C. trifolii* and *L. chartarum*, respectively. After assembly we yielded a 41.68 Mb and 37.95 Mb nuclear genome for *C. trifolii* and *L. chartarum,* respectively. Furthermore, the contig N50 was 400.60 Kb, and the scaffold N50 was 638.94 Kb for the *C. trifolii* genome, while the contig and scaffold N50 values were 234.84 Kb and 284.12 Kb for the *L. chartarum* genome, respectively. The average GC content of these two species was determined to be 49.74% and 50.64% (Table 1), respectively. Moreover, we obtained 13,649 and 15,091 protein-coding genes for these two species. Most genes in the *C. trifolii* and *L. chartarum* genomes contained few introns, with an average of 1.92 and 1.83 introns per gene, respectively. The coding sequences of the *C. trifolii* and *L. chartarum* genome showed average lengths of 496.99 bp and 506.53 bp, respectively. 

To further determine the functions of the protein-coding genes in *C. trifolii* and *L. chartarum*, the predicted protein sequences were compared to several public databases (NR, Gene Ontology (GO)), and KEGG). We identified 11,581 (84.85%) genes with a minimum of one hit with proteins in the NR database for *C. trifolii* and 11,681 (77.70%) genes annotated in the NR database for *L. chartarum*. Based on the GO analysis results for the *C. trifolii* and *L. chartarum* protein-coding genes, we assigned a total of 5,421 protein-coding genes to different GO categories for *C. trifolii* and 6,831 protein-coding genes to GO categories for *L. chartarum*. Their GO term classification was highly similar for these species (Appendix A). KEGG analysis revealed that a total of 403 and 401 KEGG metabolic pathways were annotated in the *L. chartarum* (Appendix A) and *C. trifolii* (Appendix A) genomes, respectively. 

### 2.3. General Features of the Newly Sequenced Mitochondrial Genomes

The complete sequence of the *L. chartarum* mitochondrial genome was mapped as a 68,926 bp circular molecule with an average GC content of 28.60%. Similarly, the *C. trifolii* mitogenome was also a typical circular 59,100 bp DNA molecule with an average GC content of 29.31% (Figure 2). In the *L. chartarum* mitogenome, we identified 38 protein-coding genes, 26 tRNA genes, and 2 rRNA genes (*rrnL* and *rrnS*) (Appendix A), which were located on both strands. Among the protein-coding genes, 12 had identified functions, while the remaining were determined to be open reading frames (ORFs). The overall GC content of the protein-coding genes was 29.78% in the *L. chartarum* mitogenome, ranging from 21.21% (ORF109) to 43.17% (ORF270), indicating that these two mitogenomes had high A+T content (Figure 3). In total, we found 13 introns located in *cob* (1) *cox1*&*2* (7), *cox3* (1) and *nad5* (2) (Appendix A) in the *L. chartarum* mitogenome. In the *C. trifolii* mitogenome, there were 28 protein-coding genes, 23 tRNA genes, and 2 rRNA genes (*rrnL* and *rrnS*) (Appendix A). In the *C. trifolii* mitogenome, the GC content of the protein-coding genes was 28.12%, ranging from 22.92% (*nad6*) to 32.23% (*cox1*) (Appendix A). We also found 11 introns distributed in the *atp6* (2), *cob* (2), *cox1* (1), *cox2* (1), and *nad1* (2) genes.

### 2.4. Genome Comparison

The nuclear genomes of *L. chartarum* and *C. trifolii* were similar in size as well as GO and KEGG annotation. To better understand the variation in mitogenome components between *L. chartarum*, *C. trifolii* and related species, a comparative analysis of the protein-coding genes was performed. Five representative species from different genera within the order Pleosporales (Pleosporaceae) were chosen: *Pyrenophora tritici-repentis*, *L. chartarum*, *C. trifolii*, *Bipolaris cookei*, and *Stemphylium lycopersici*, and compared for gene content and genome collinearity (Figure 4).

### 2.5. Evolutionary Rates of Nuclear and Mitochondrial Genes

To detect the nature of the evolutionary selection pressure in *L. chartarum* and *C. trifolii*, seven and 12 nuclear and mitochondrial genes were chosen, respectively, to calculate Ka, Ks and Ka/Ks values. The results showed that the Ka/Ks ratio ranged from 0.06–3.11 for the nuclear genes and from 0.04–1.15 for the mitochondrial genes, suggesting that these genes underwent negative selection pressure in the evolution process. Meanwhile, the average Ka/Ks values of the nuclear and mitochondrial genes were 0.99 and 0.26, respectively. The highest Ka/Ks ratio in nuclear genes was found in *RPB1* (3.11), followed by *EF* (2.84) and *ITS* (0.71). The highest Ka/Ks ratio in mitochondrial genes was found in *nad3* (1.15), followed by *cob* (0.75) and *cox3* (0.27) (Figure 5A,B). Our studies showed that genes such as *EF*, *RPB1* and *nad3* underwent positive selection, while other genes, notably the mitochondrial genes, underwent pure selection during their evolutionary process. 

### 2.6. Phylogenetic Relationship 

To acquire additional evidence for the classification of Pleosporales species, and to understand the evolutionary history of the mitochondrial genome, we conducted separate phylogenetic analyses for individual genes as well as for the combined nuclear and mitochondrial DNA datasets (Figure 6 and Figure 7). Differences were observed in the topologies of the phylogenetic trees generated by individual nuclear genes and mitochondrial genes (Appendix A). However, their evolutionary relationships appeared to be interdependent. For the phylogenetic relationship inferred based on the combined nuclear genes, the results showed that *C. trifolii* were found to be sister groups of *B. cookei*, *P. tritici-repentis* and *S. lycopersici* with high confidence (Figure 6). Moreover, this group clustered with clades containing species in *Leptosphaeria*, and *Parastagonospora*. *M. graminicola* was located in the basal position of the phylogenetic tree. Similarly, the phylogenetic tree inferred from mitochondrial genomes showed that *C. trifolii* clustered with *S. lycopersici*, *P. tritici-repentis* and *B. cookei*, which clustered with the clade containing *P. nodorunm* and *Skeletocutis bambusicola* (Figure 7). We found that two species in Leptosphaeriaceae were clustered together, and *L. maculans* was located in the medium position of these two clades. It appeared that *Didymella pinodes* is at a greater genetic distance from other species in the Pleosporales order.

## 3. Discussion

Fungal endophytes are widely distributed in a great variety of plant tissues [34]. We found that symbiont endophytic fungi are dispersed throughout all tissues of tobacco plants and play important roles in hastening growth, reducing heavy metal content, improving plant protection, etc. Therefore, studies were performed to determine the species characteristics and evolutionary position of endophytic fungi in tobacco. We isolated two particularly important parasitic species, *L. chartarum* and *C. trifolii*, that cause leaf disease on different type of feed crops [6], causing economic loss. The assembled genomes of *C. trifolii* and *L. chartarum* are 41.68 Mb and 37.95 Mb in size, respectively, which are larger than the previously published an unclassified Pleosporales UM1110 ] [35], *Shiraia bambusicola* [36], and *Stagonospora nodorum* (36.91 Mb) and *Cochliobolus heterostrophus C5* (36.19 Mb) [37,38], although smaller that of the species in the Dothideomycetes fungi class, such as *Mycosphaerella fijiensis*, which has an assembled genome size of 63.95 Mb.. In addition, we found that the percentage of repeat elements contributed to the genome size variation among these species. The assembled genome had a GC content approximately 50%, which is near to that in other species ranging from 45.55%–52.33% [37,38].

Both the GC content and gene content of the complete mitogenomes of *L. chartarum* and *C. trifolii* are comparable to previously reported mitogenomes of species belonging to the Pleosporales [33,38]. The mitochondrial genome codon usage in *L. chartarum* and *C. trifolii* showed a significant bias toward A and T, as is also reported for other fungal species [38,39]. In the *L. chartarum* and *C. trifolii* mitochondrial genomes, Ile, Leu, Lys, and Phe are the most frequently encoded amino acids. We found that UUU, AUU, AAU, and UAU are the most frequent codons in both of these genomes. Since these frequently used codons are exclusively comprised of A and T (U) (Figure 2), they contribute to the high A + T content seen in most fungal mitochondrial genomes, including that of *Beauveria pseudobassiana* and *Madurella mycetomatis* [40,41]. Furthermore, all tRNAs and rRNAs from these newly sequenced mitochondrial genomes are comparable to those in related species [42,43]. This preferred codon usage is strongly reflected at the third position by the high A/T versus G/C frequencies, which is consistent with the rich A+T content in the whole mitogenomes. 

The genome sizes of these mitogenomes are larger than those of closely related species *Shiraia bambusicola* (39,030 bp), *Phaeosphaeria nodorunm* (49,761bp) and *Bipolaris maydis* (37,250 bp), while being much smaller than the mitogenomes of *Leptosphaeria maculans* (154,863bp) and *P. tritici-repentis* (157,011bp). At least one group I or group II introns are located within most fungal mitogenomes. For example, there are 39 introns in the *Pezizomycotina subphylum* mitogenome and two in the *Fusarium oxysporum* mitogenome [44]. Moreover, 1, 5, 5, 13, 15, 18, 33, 35, 38 introns were identified in *S. bambusicola, B. maydis*, *P. nodorum*, *S. lycopersici*, *Didymella pinodes*, *P.tritici-repentis*, *B. cookie*, *Leptosphaeria biglobosa*, *Leptosphaeria maculans*, respectively. However, we did not identify any introns within the *M. graminicola* mitogenome, which was consistent with a previous study [31]. Moreover, we observed a positive correlation between the number of introns and genome size among Pleosporales species. In addition, we found that *P. tritici-repentis* contains 79 ORFs, while there are only 5 ORFs in *S. bambusicola*. Therefore, we speculated differences in the number of ORFs and the presence of introns may be attributed to the various sizes of the fungi mitogenomes, which is supported by previous studies [45,46].

We also determined that the nuclear genomes of *L. chartarum* and *C. trifolii* are similar in size and annotation, which is also similar to other Pleosporales species [47,48]. The results from mitochondrial genome analysis showed that the gene content is conserved in these species as their genome size ranged from 37,250–157,001 bp [33,49]. Further, the genome collinearity analysis showed gene content frequency, and frequent gene structure rearrangement events, even among species in the same class. In addition, we found the Ka/Ks ratios of genes such as *EF*, *RPB1* and *nad3* were more than one, indicating that they underwent positive selection, while other genes underwent pure selection (Ka/Ks < 1). Previous studies also reported Ka/Ks ratios for nuclear orthologous pairs as higher than one, while the Ka/Ks values for protein-coding genes in mitochondrial genomes were found to be less than one [50,51], which was similar to what we observed. These results suggest that the evolution rates of nuclear and mitochondrial genes differed during the evolution process.

The results also confirm that the phylogenetic trees created by combining nuclear and mitochondrial DNA datasets are consistent with those from previous studies [25], although they possess minor differences in topologies. However, we found it difficult to obtain consistent results in the phylogenetic trees generated by single gene analysis regardless of whether single nuclear or mitochondrial genes were applied. In our opinion, the individual genes are undergoing rapid evolution although they harbor different evolutionary rates, making it difficult to observe accurate phylogenetic relationships between taxa using only single genes for identification of endophytic fungi species. Therefore, aligned datasets based on multi-genes or the whole genome more accurately represent phylogenetic relationships than those constructed using a single gene [52].

Currently, mitogenomes are commonly selected as molecular markers for phylogenetic and evolutionary analysis as they are semi-autonomous organelles with conservative gene contents and a stable gene structure [33,43,44,45,46]. However, as mentioned previously, only a limited number of mitogenomes have been published for the Pleosporales species, even in the class Dothideomycetes [23,25,36]. Therefore, our phylogenetic tree generated from whole mitochondrial genomes serves to further the current understanding on phylogenetic relationships between Pleosporales species. Although previous studies have successfully identified species and performed phylogenetic analysis based on single genes [4,6,8,23]. Currently, it has become increasingly common to use multiple genes to reconstruct phylogenies, as individual genes may experience different adaption pressures and, thus, produce unique substitution parameters. Compared with the phylogenetic tree generated from single genes, that constructed from mitogenomes was more accurate in determining evolutionary position confirmation for each organism. However, we also observed significant differences in mitogenome size, ORF number and intron number between species, which makes it challenging to construct their phylogenetic relationships using the entire genome or non-coding regions. The symbiont endophytic fungi belonging to the order Pleosporales are highly diverse [33,49], requiring effective and accurate ways to infer their relationships. Herein, we confirm that it is more accurate to examine phylogenetic relationships of endophytic fungi in tobacco via analysis of datasets comprised of multi-protein-coding genes from nuclear or mitochondrial genomes.

The information of genomes (nuclear and mitogenomes) of *L. chartarum* and *C. trifolii* and their phylogenetic positions can help us understand their features, and can help us explore their secondary metabolites. After having a clear understanding of their genomes and phylogenetic position, we are committed to studying the interaction mechanisms between these endophytic fungi and their host tobacco plants. Like most endophytes, endophytic fungi in tobacco have great merits including promoting plant growth, improving resistance to heavy metals, bacteriostasis, etc. [28]. The endophytic fungi perform their functions primarily through their secondary metabolites. Hence, increasing secondary metabolites were isolated and purified and then were used to evaluate their biological functions. Currently, we have isolated some secondary metabolites, such as prenylated indole alkaloids (PIAs), 11-α-methoxycurvularin, sesquiterpenes, etc., and some novel types from these two fungi. In future, we will detect the biological activities of these secondary metabolites, such as anti-cytotoxicity and antibacterial activity, etc., in order to promote tobacco growth and prevent and control tobacco diseases. Therefore, an understanding of these two endophytic fungi genome characteristics and evolutionary histories could not only contribute to explore the secondary metabolites in tobacco, but provide a deeper understanding of host–endophyte relationships at the molecular level.

## 4. Materials and Methods

### 4.1. Materials, Isolation, and Culture

The endophytic fungus *C. trifolii* was first isolated from healthy leaves of *Nicotiana tabacum* L., which were collected from Enshi (108°23′12″–110°38′08″E, 29°07′10″–31°24′13″N), Hubei province, P. R. China, in July 2016. The fungal strain *L. chartarum* was isolated from the fresh stems of *Nicotiana tabacum*, also collected from Enshi, Hubei. The fungi were identified by analysis of their internal transcribed spacer (ITS), V1, and V4 regions of rDNA, as described in our previous report [53], as well as by their morphologies. Both strains are currently deposited at the Tobacco Research Institute of the Chinese Academy of Agricultural Sciences. In addition, the phylogenetic analysis of *C. trifolii* and *L. chartarum* was performed using MEGA 6.0 software MEGA version 6.0 (Koichiro Tamura, Hachioji, Tokyo, Japan) [54].

### 4.2. DNA Extraction

After isolation and purification, the hyphae of *C. trifolii* and *L. chartarum* were scraped from the surface of the agar and ground in the presence of liquid nitrogen to form a fine powder. Genomic DNA was then extracted using a fungal DNA extraction kit (Omega Bio-tek, Norcross, GA, USA), following the manufacturer’s instructions. Electrophoresis on 1% agarose gels was used to assess the integrity of DNA samples and the purity and concentration were determined using a Nanodrop microspectrophotometer (ND-2000, Thermo Fisher Scientific, Waltham, MA, USA) and a Qubit 2.0. fluorometer (Life Technologies, Carlsbad, CA, USA), respectively. Genomic DNA was stored at −20 °C prior to library construction.

### 4.3. Nuclear and Mitochondrial Genome Assembly

Approximately 2 μg of genomic DNA from each strain was fragmented into 500 bp fragments, followed by construction of an Illumina genome sequencing library using the NEBNext Ultra II DNA Library Prep Kits (NEB, Beijing, China) according to the manufacturer’s instructions. These libraries were then sequenced on an Illumina HiSeq 2500 Platform (Illumina, San Diego, CA, USA). Briefly, raw sequence reads were first quality trimmed and adapter sequences were removed using FastQC (www.bioinformatics.babraham.ac.uk/projects/fastqc/) [55]. The sequences were deposited in DRYAD (https://doi.org/10.5061/dryad.wh70rxwjq). Subsequently, the SPAdes 3.9.0 software was used for the *de novo* assembly of the species [56]. 

For the mitochondrial genome assembly, the assembled contigs were annotated using BLAST (http://blast.ncbi.nlm.nih.gov/; conditions: query coverage ≥m70%; E-value ≤ 1e-10). Next, the mitochondrial-related contigs of these two species were screened out, and the orientation of the contigs was confirmed based on published data for other Pleosporales species. MITObim V1.9 was then used to close the gaps between contigs [56,57]. In addition, the accuracy of these sequences was confirmed by iterative mitochondrial baiting [57]. 

### 4.4. Genome Prediction, Annotation and Analysis

Gene predictions for the masked genome of *C. trifolii* and *L. chartarum* were performed using AUGUSTUS [58]. Next, gene annotation, Nr BLAST was performed locally using BlastP with an e-value of e^−5^, and GO was assigned using the results from Blast2GO [59]. The protein-coding genes were then searched against the KEGG database (http://www.genome.jp/kegg) to perform annotations for the whole genome [60].

To annotate these two complete mitogenomes, MFannot (http://megasun.bch.umontreal.ca/cgi-bin/mfannot/mfannotInterface.pl) was performed. Following this, tRNAs were identified using tRNAscan-SE 1.21 search server (http://lowelab.ucsc.edu/tRNAscan-SE/) with default settings [61]. rRNA genes were also identified using multi-sequence alignment with the GenBank sequences of *B. cookie* and *P. chartarum* mitogenomes as reference using BLAST (http://blast.ncbi.nlm.nih.gov/Blast.cgi) searches. The base composition of these two mitogenomes were analyzed with DNAStar Lasergene v7.1 (http://www.dnastar.com/). In addition, AT skews and GC skews were determined using (A%-T%)/(A%+T%) and (G%-C%)/(G%+C%), respectively, to characterize strand asymmetries in the mitogenomes. CodonW was used to analyze codon usage frequency for amino acids in the genomes [62]. The graphical maps of the mitogenomes were constructed using OGDRAW (http://ogdraw.mpimp-golm.mpg.de/cgi-bin/ogdraw.pl) [56]. A comparative analysis of nucleotide sequences for each protein-coding gene, and two ribosomal DNA genes among *Pyrenophora tritici-repentis*, *L. chartarum*, *C. trifolii*, *Bipolaris cookei*, *Stemphylium lycopersici* was conducted. Mitogenomes were also annotated using MFannot.

### 4.5. Evolutionary Rates of the Nuclear Genes and Mitochondrial Genes

The synonymous sites (dS) and the non-synonymous substitution sites (dN) as well as their ratios (dN/dS) are often employed to measure evolutionary rates. Therefore, seven nuclear genes (factor-1 alpha (*TEF1*), largest and second-largest subunits of DNA-directed RNA polymerase (*RPB1, RPB2*), β-tubulin (*BT2*), β-actin, glyceraldehyde-3-phosphate dehydrogenase (*GAPDH*), nuclear rDNA internal transcribed spacers (*ITS*)) and 12 mitochondrial genes (*atp6*, *cob*, *cox1*, *cox2*, *cox3*, *nad1*, *nad2*, *nad3*, *nad4*, *nad4L*, *nad5*, and *nad6*) were chosen to calculate values of Ka, Ks and Ka/Ks. First, these genes were aligned using MEGA 6.0 according to codons (parameters: Gap opening penalty:10; Gap extension penalty:0.2; Delay divergent cutoff:30%). Subsequently, we calculated the ratio of dN/dS using DnaSP ver. 5 [63]. 

### 4.6. Phylogenetic Analysis

To determine the phylogenetic relationships of *C. trifolii* and *L. chartarum*, two datasets for the nuclear genome and mitochondrial genome were selected. Since the class Dothideomycetes includes the orders Botryosphaeriales, Pleosporales, and Capnodiales, *Mycosphaerella graminicola* (Capnodiales) was selected as an outgroup when the phylogenetic tree was constructed.

For the nuclear genome, the seven genes mentioned above were selected according to the above selection criteria for constructing the tree. These genes were aligned by MAFFT using default settings and then concatenated head-to-tail to form the final datasets [64]. The phylogenetic tree was then reconstructed using RAxML version 8.1.12 [65] and MrBayes [66].

For mitogenomes, construction of the phylogenetic tree was performed using a combined dataset consisting of the 12 common protein-coding genes that are encoded in *C. trifolii* and *L. chartarum,* and other related mitochondrial genomes. These genes were also aligned and used to construct the phylogenetic tree according to the previously described method [64,65,66]. Nucleotide substitution models of the final datasets were selected using jModelTest [67]. For each node of the maximum likelihood (ML) tree, the bootstrap support was calculated using 1,000 replicates. For the Bayesian tree, the initial 10% of trees was ruled out burn-in, and four simultaneous chains were run for 10,000,000 generations.

## 5. Conclusions

Fungal endophytes in tobacco are widely distributed in nearly all tissues where they participate in a myriad of biological functions. Therefore, it is important to determine the species distribution and characteristics of endophytic fungi in tobacco plants. Using healthy tissues from tobacco as the source, we isolated and identified two endophytic fungi, namely *L. chartarum* and *C. trifolii,* and obtained their nuclear mitogenome sequences. We successfully annotated their protein-coding genes, and constructed a corresponding phylogenetic relationship using data for other available Pleosporales species. These data provide us with new insights for determining the species distribution and characteristics of endophytic fungi in tobacco.

## Figures and Tables

**Figure 1 ijms-21-02461-f001:**
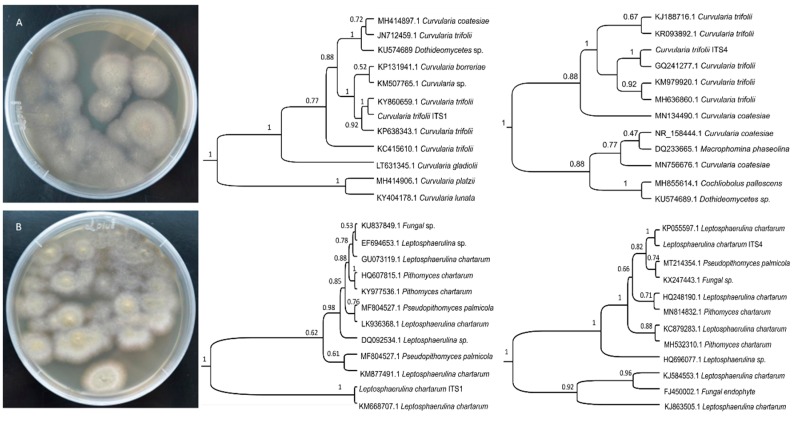
The cultural morphology of *C. trifolii* (**A**) and *L. chartarum* (**B**), and their phylogenetic relationships on the basis of ITS regions.

**Figure 2 ijms-21-02461-f002:**
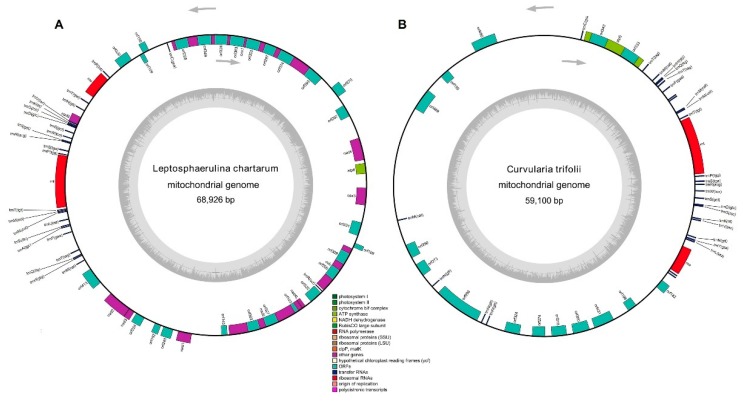
Circular mapping of the complete mitochondrial genome from (**A**) *L. chartarum* and (**B**) *C. trifolii*. The two mitogenomes were annotated using MFannot and were used to construct their graphical maps using OGDRAW (http://ogdraw.mpimp-golm.mpg.de/cgi-bin/ogdraw.pl).

**Figure 3 ijms-21-02461-f003:**
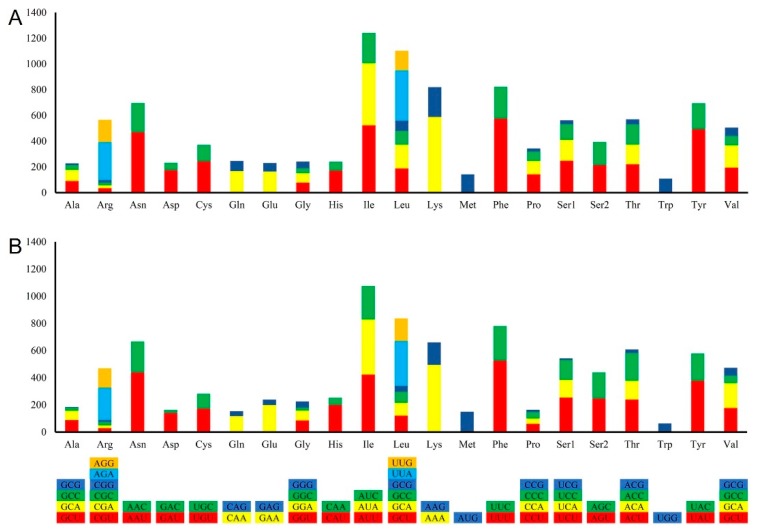
Mitochondrial genome codon usage in (**A**) *C. trifolii* and (**B**) *L. chartarum*. CodonW was used to analyze codon usage frequency for amino acids in these two genomes. Boxes with different colors represent different codon usage.

**Figure 4 ijms-21-02461-f004:**
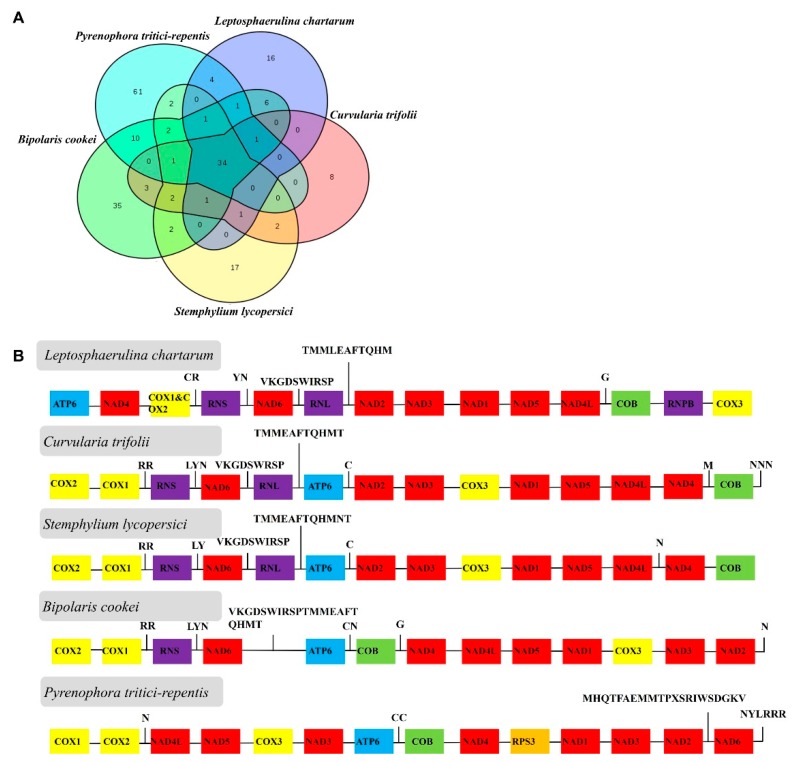
Comparison of the mitogenome components in *L. chartarum*, *C. trifolii* and related species. (**A**) Gene content comparison among *Pyrenophora tritici-repentis*, *L. chartarum*, *C. trifolii*, *Bipolaris cookei*, *Stemphylium lycopersici* mitogenomes. (**B**) Comparative analysis of nucleotide sequences for each protein-coding gene, two ribosomal DNA genes among *Pyrenophora tritici-repentis*, *L. chartarum*, *C. trifolii*, *Bipolaris cookei*, *Stemphylium lycopersici.* The same genes are marked with the same color.

**Figure 5 ijms-21-02461-f005:**
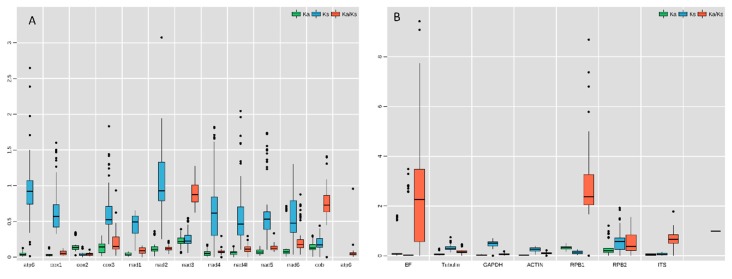
Evolutionary rates of the mitochondrial genes (**A**) and nuclear genes (**B**). The Ka (non-synonymous) and Ks (synonymous) values, as well as their Ka/Ks ratio appear as green, blue and red box plots, respectively. The bar represents standard deviation (SD), median and epitomizes values between the first quartile and the third quartile of Ka, Ks and Ka/Ks.

**Figure 6 ijms-21-02461-f006:**
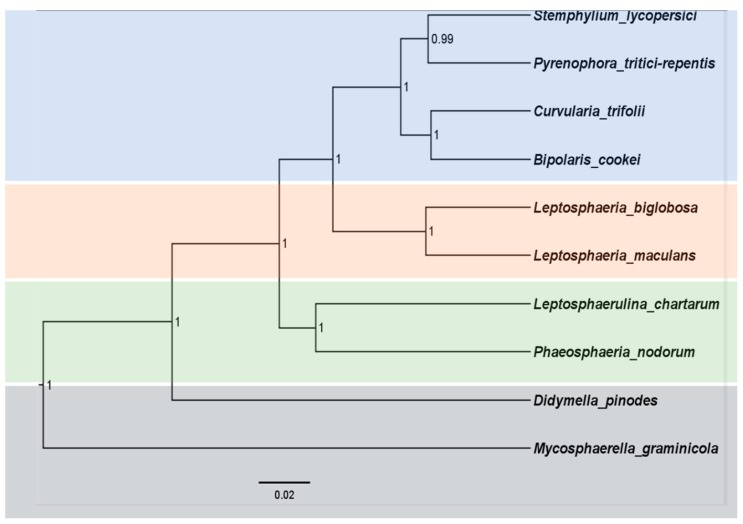
Phylogenetic analyses for combined nuclear DNA datasets. A combined dataset consisting of the seven nuclear genes (*TEF1*, *RPB1, RPB2*, *BT2*, β-actin, *GAPDH*, *ITS* were used to construct this phylogenetic tree.

**Figure 7 ijms-21-02461-f007:**
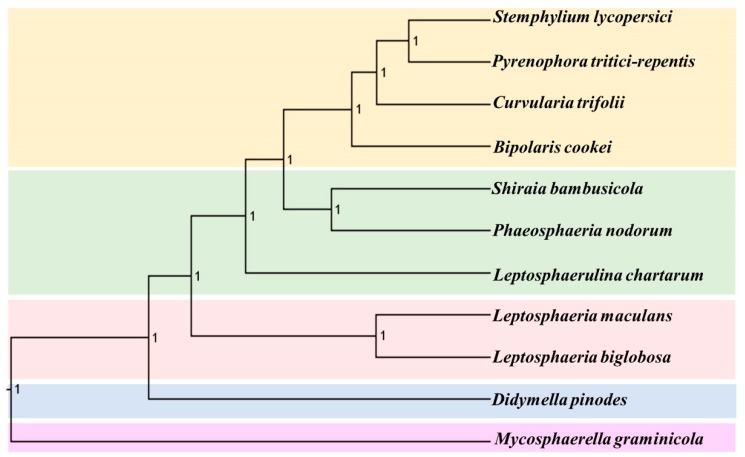
Phylogenetic analyses for combined mitochondrial DNA datasets. A combined dataset consisting of the 12 common protein-coding genes (*atp6*, *cob*, *cox1*, *cox2*, *cox3*, *nad1*, *nad2*, *nad3*, *nad4*, *nad4L*, *nad5*, and *nad6*) that are encoded in *C. trifolii* and *L. chartarum* and other related mitochondrial genomes was used to construct this phylogenetic tree.

**Table 1 ijms-21-02461-t001:** Statistics of the final assembly of *L. chartarum* and *C. trifolii* genome.

	*L. chartarum*		*C. trifolii*	
Sample ID	Contig	Scaffold	Contig	Scaffold
Total(bp)	37,945,625	37,947,115	41,681,125	41,682,726
Max(bp)	1,026,975	1,409,484	1,476,666	1,819,210
N50	234,835	284,119	400,595	638,944
N90	76,366	94,298	49,549	65,120
GC content	50.65 %	50.64%	49.75 %	49.74 %
Ns	0.00%	0.01%	0.00%	0.01%

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
