# Peer review of "Characterization of Nuclear and Mitochondrial Genomes of Two Tobacco Endophytic Fungi Leptosphaerulina chartarum and Curvularia trifolii and Their Contributions to Phylogenetic Implications in the Pleosporales"

_ijms, 2020, doi:10.3390/ijms21072461_

Round 1

Reviewer 1 Report

The authors isolated and sequenced two endophytic fungal species of tobacco plants: Curvularia trifolii and Leptosphaerulina chartarum. Their nuclear and mitochondrial genomes were sequenced, and phylogenetic analyses were carried out on the base of 7 and 12 nuclear genes respectively. The results are interesting, but the meaning of many sentences, especially in the discussion paragraph, is too difficult to understand due a very poor English style. Despite the number of molecular studies carried out and the interesting conclusions of the work, the paper lack of rigor and clarity. It is often inaccurate and approximate; many sentences are unclear. English is very, very poor and needs to be deeply implemented. 

Author Response

Comments and Suggestions for Authors

The authors isolated and sequenced two endophytic fungal species of tobacco plants: Curvularia trifolii and Leptosphaerulina chartarum. Their nuclear and mitochondrial genomes were sequenced, and phylogenetic analyses were carried out on the base of 7 and 12 nuclear genes respectively. The results are interesting, but the meaning of many sentences, especially in the discussion paragraph, is too difficult to understand due a very poor English style. Despite the number of molecular studies carried out and the interesting conclusions of the work, the paper lack of rigor and clarity. It is often inaccurate and approximate; many sentences are unclear. English is very, very poor and needs to be deeply implemented.

Response: Thank you very much for your evaluation and comments on our manuscript, especially for the detailed changes. Now, we have revised our manuscript according to your suggestions, also with the language, grammatical mistakes etc. For example, we have asked a professional organization to help us polish our English. Meanwhile, the Figure 1, 2, and 3 also revised according to your suggestion. The other revised details can be found in our manuscript.

Reviewer 2 Report

The manuscript is well written.  Experimental plan is well designed.  I have only one question to raise: How the isolated endophityc fungi interfere with tobacco tissue function.  Authors shouls discuss this issue in the text (Introduction and/or discussion).

Author Response

Reviewer 2

Comments and Suggestions for Authors

The manuscript is well written. Experimental plan is well designed.  I have only one question to raise: How the isolated endophityc fungi interfere with tobacco tissue function.  Authors should discuss this issue in the text (Introduction and/or discussion).

Response: Thanks for your suggestion. We have added some description in the introduction and discussion sections. The details can be found in our manuscript.

Round 2

Reviewer 1 Report

The manus has been deeply improved, I greatly appreciated your revision. I read it with interest and  inserted some minor corrections.

Author Response

Dear reviewer:

We are very grateful for your suggestions, and we have already read
your comments carefully and then made changes accordingly.  All modifications in the revised manuscript were marked in red.